# Adjuvant PD-1 and PD-L1 Inhibitors and Relapse-Free Survival in Cancer Patients: The MOUSEION-04 Study

**DOI:** 10.3390/cancers14174142

**Published:** 2022-08-26

**Authors:** Alessandro Rizzo, Veronica Mollica, Andrea Marchetti, Giacomo Nuvola, Matteo Rosellini, Elisa Tassinari, Javier Molina-Cerrillo, Zin W. Myint, Tomas Buchler, Fernando Sabino Marques Monteiro, Enrique Grande, Matteo Santoni, Francesco Massari

**Affiliations:** 1Struttura Semplice Dipartimentale di Oncologia Medica per la Presa in Carico Globale del Paziente Oncologico “Don Tonino Bello”, Istituto di Ricerca e Cura a Carattere Scientifico (IRCCS), Istituto Tumori Giovanni Paolo II-Bari, Viale Orazio Flacco 65, 70124 Bari, Italy; 2Medical Oncology, IRCCS Azienda Ospedaliero-Universitaria di Bologna, 40138 Bologna, Italy; 3Department of Experimental, Diagnostic and Specialty Medicine, S.Orsola-Malpighi University Hospital, University of Bologna, 40138 Bologna, Italy; 4Department of Medical Oncology, Hospital Ramón y Cajal, 28034 Madrid, Spain; 5Markey Cancer Center, University of Kentucky, Lexington, KY 40536-0293, USA; 6Department of Oncology, First Faculty of Medicine, Charles University and Thomayer University Hospital, 14059 Prague, Czech Republic; 7Hospital Santa Lucia, Brasilia 70390-700, Brazil; 8Hospital Universitário de Brasilia, Brasilia 70840-901, Brazil; 9Latin American Cooperative Oncology Group-LACOG, Porto Alegre 90619-900, Brazil; 10Department of Medical Oncology, MD Anderson Cancer Center Madrid, 28033 Madrid, Spain; 11Oncology Unit, Macerata Hospital, via Santa Lucia 2, 62100 Macerata, Italy

**Keywords:** adjuvant, immunotherapy, immune checkpoint inhibitors, PD-1, relapse-free survival

## Abstract

**Simple Summary:**

Despite a significant improvement in clinical outcomes and the emergence of novel and potentially curative strategies, a noticeable number of oncological patients witness a disease relapse after surgery. Adjuvant treatments have been developed to reduce the risk of recurrence and gain survival benefits for these patients. The aim of this meta-analysis was to explore the impact of adjuvant PD-1/PD-L1 inhibitors on relapse-free survival in cancer patients with many solid tumors. We confirmed that PD-1/PD-L1 inhibitors may reduce the risk of relapse in many tumor types, compared to control treatments. Moreover, we showed that the benefit was consistent in subgroups divided according to gender and age.

**Abstract:**

Background: Adjuvant treatment has always been a cornerstone in the therapeutic approach of many cancers, considering its role in reducing the risk of relapse and, in some cases, increasing overall survival. Adjuvant immune checkpoint inhibitors have been tested in different malignancies. Methods: We performed a meta-analysis aimed to explore the impact of adjuvant PD-1 and PD-L1 inhibitors on relapse-free survival (RFS) in cancer patients enrolled in randomized controlled clinical trials. We retrieved all phase III trials published from 15 June 2008 to 15 May 2022, evaluating PD-1/PD-L1 inhibitors monotherapy as an adjuvant treatment by searching on EMBASE, Cochrane Library, and PubMed/ Medline, and international oncological meetings’ abstracts. The outcome of interest was RFS. We also performed subgroup analyses focused on age and gender. Results: Overall, 8 studies, involving more than 6000 patients, were included in the analysis. The pooled results highlighted that the use of adjuvant PD-1/PD-L1 inhibitors may reduce the risk of relapse compared to control treatments (hazard ratio, 0.72; 95% confidence intervals, 0.67–0.78). In addition, the subgroup analyses observed that this benefit was consistent in different patient populations, including male, female, younger, and older patients. Conclusions: Adjuvant anti-PD-1/PD-L1 treatment is associated with an increased RFS in the overall population and in subgroups divided according to age and gender.

## 1. Introduction

Despite the fact that the last two decades have seen the improvement of clinical outcomes and the emergence of novel, potentially curative treatment options, a notable proportion of cancer patients experience disease relapse after radical surgery. As a result, adjuvant anticancer therapies have been developed to lower the risk for recurrence and to improve overall survival. Among these, recent years have seen the emergence of anticancer immunotherapy, which represents a standard for the management of a wide spectrum of metastatic solid tumors, ranging from malignant melanoma to non-small cell lung cancer (NSCLC), hepatocellular carcinoma (HCC), renal cell carcinoma (RCC), and urothelial carcinoma (UC) [1,2,3,4,5]. Immune checkpoint inhibitors (ICIs) target the immune checkpoints on T-lymphocytes and regulate the function of the anticancer immune response through molecular mechanisms [6,7]; cytotoxic T lymphocyte antigen-4 (CTLA-4), programmed cell death-1 (PD-1), and its ligands PD-L1 and PD-L2 represent the most frequently used ICIs target, something that has led to the development of specific inhibitors, such as—among others—pembrolizumab, nivolumab, durvalumab, atezolizumab, and avelumab [8]. Following the practice-changing results observed in clinical trials evaluating ICIs, these agents have been assessed and are currently under investigation in the earlier stages of disease, including adjuvant treatment [9,10,11,12,13,14,15,16]. Adjuvant immunotherapy is administered following resection surgery, with ICIs increasing the frequency of activated T cells that have been suggested to be able to eliminate tumor cells after radical resection [17,18]. However, the use of immunotherapy in the adjuvant setting raises several questions, including the duration of treatment, the selection of appropriate comparators; the sequencing of therapies (especially in tumors with targetable mutations, such as melanoma); and the safety profile of these agents, a key point to consider after potentially curative surgery. At the same time, several trials with available data lack overall survival results and a mature follow-up.

Based on these premises, we performed a comprehensive and up-to-date meta-analysis aiming to evaluate the impact of adjuvant PD-1 and PD-L1 inhibitors on relapse-free survival (RFS) in cancer patients. Subgroup analyses exploring the role of gender and age were also performed.

## 2. Materials and Methods

### 2.1. Search Strategy

All phase III clinical trials published from June 15th, 2008, to May 15th, 2022, assessing anti-PD-1 and anti-PD-L1 agents’ monotherapy as adjuvant treatment in cancer patients, were retrieved by three different authors. Keywords used for searching on EMBASE, Cochrane Library, and PubMed/ Medline were as follows: “cancer” OR “solid tumor” AND “atezolizumab” OR “avelumab” OR “durvalumab” OR “immune checkpoint inhibitor” OR “nivolumab” OR “PD-1” OR “PD-L1” OR “pembrolizumab” OR “Programmed death receptor-1” OR “immunotherapy”. Articles published in peer-reviewed journals and written in English language were included, and the proceedings of the main meetings were also searched for relevant reports.

### 2.2. Selection Criteria

Trials retrieved from the first analysis we conducted were subsequently restricted to: (1) prospective phase III clinical trials in cancer patients; (2) participants receiving adjuvant treatment with PD-1 or PD-L1 inhibitors; (3) studies with available data in terms of RFS; and (4) studies with available data in male, female, elderly, and younger patients.

### 2.3. Data Extraction

The following data were extracted for each publication: (1) general trial information; (2) treatment arms; (3) the number of cancer patients; and (4) the available outcomes in terms of RFS in patients treated with anti-PD-1 or PD-L1 agents. The analysis was conducted according to Preferred Reporting Items for Systematic Review and Meta-Analyses (PRISMA) guidelines (Appendix A) [19].

### 2.4. Risk of Bias Assessment in Included Studies

The Cochrane Collaboration tool was used to evaluate the methodological quality of the included studies; a risk of bias in the selected studies was assessed independently by three separate authors [20]. Trials examined were graded as having a “low risk”, “high risk”, or “unclear risk” of bias across the specified domains of selection, performance, attrition, and reporting bias. The results of the assessment were summarized in a risk of bias graph (Figure 1). The presence of publication bias was formally evaluated using funnel plots (Appendix A).

### 2.5. Statistical Design

ProMeta 3 software was used to perform all the statistical analyses. Hazard ratios (HRs) and 95% confidence intervals (CIs) were the effect measures for RFS, and these values were extracted from available studies. Forest plots were used to assess HRs to describe the relationship between treatment and RFS in the specified cohorts of patients. Statistical heterogeneity between trials was examined using the Chi-square test and the I2 statistic; substantial heterogeneity was considered to exist when the I2 value was greater than 50% or there was a low *p* value (<0.10) in the Chi-square test [21]. We applied the fixed effects model when no heterogeneity was noted, while the random effects model was used in the case of significant heterogeneity.

## 3. Results

### 3.1. Selected Studies

2395 potentially relevant reports were identified; these reports were later restricted to eight following the independent evaluation of three authors [9,10,11,12,13,14,15,16]; 2387 records were excluded as non-pertinent reports. Eligible studies were identified and selected as shown in Figure 2; a summary of the included trials is presented in Table 1 [9,10,11,12,13,14,15,16]. The eight studies included in the meta-analysis compared adjuvant PD-1 or PD-L1 inhibitors monotherapy in cancer patients, involving more than 6000 patients [9,10,11,12,13,14,15,16]. In particular, these studies included patients with melanoma, UC, esophageal or gastroesophageal junction cancer, NSCLC, and RCC. Experimental treatment included atezolizumab, nivolumab, and pembrolizumab.

### 3.2. Relapse Free Ssurvival

The pooled HR for RFS was 0.72 (95% CI, 0.67–0.78), suggesting that patients receiving adjuvant PD-1 or PD-L1 inhibitors presented longer RFS (Figure 3); the analysis was associated with low heterogeneity (I2 of 31%); thus, a fixed-effects model was used.

### 3.3. Relapse-Free Survival According to Gender

The pooled HRs for RFS in male and female patients were 0.74 (95% CI, 0.67–0.81) and 0.73 (95% CI, 0.63–0.84) (Figure 4, Figure 5); the two analyses reported low heterogeneity (I2 of 22 and 28%, respectively), and thus a fixed-effects model was used.

### 3.4. Relapse-Free Survival According to Age

The pooled HRs for RFS in younger (<65 years) and elderly (equal or more than 65 years) cancer patients were 0.70 (95% CI, 0.63–0.77) and 0.81 (95% CI, 0.71–0.92) (Figure 6, Figure 7); the two analyses reported low heterogeneity (I2 of 17 and 19%, respectively), and thus a fixed-effects model was used.

### 3.5. Publication Bias

The funnel plots of RFS in trials comparing adjuvant PD-1 and PD-L1 inhibitors versus control treatments included in our analysis showed basic symmetric, suggesting no publication bias (Appendix A).

## 4. Discussion

Adjuvant ICIs aim to eliminate residual microscopic tumor cells via the immune system, in order to reduce the risk of cancer recurrence and to improve the chance of cure [22,23]. However, only some solid tumors benefit from this treatment approach, and better knowledge of the mechanisms leading to tumor escape and the impact of factors such as tumor microenvironment on the efficacy of immunotherapy is considered crucial for the optimal management of adjuvant ICIs [24,25]. In fact, a strong, relevant interplay between the tumor microenvironment and immune response has been highlighted in recent years, and this relationship may be particularly important in this setting. As in the case of other anticancer agents, some reflections regarding the efficacy of adjuvant ICIs in only some tumors come to mind. A question may be if we are treating the wrong patients or if we are using the wrong drugs. As regards the first point, the identification of cancer patients that are at particularly high risk for recurrence, and which may benefit from adjuvant ICIs, is a key challenge. Similarly, if clinicians and researchers are testing and using the wrong drugs, another possible challenge would be to identify agents able to achieve higher objective response rates. These two points may be both fundamental in the current and future study design of adjuvant immunotherapy.

Herein, we performed a meta-analysis aimed to explore the impact of adjuvant PD-1 and PD-L1 inhibitors on RFS in cancer patients included in randomized controlled clinical trials. To the best of the authors’ knowledge, the current study represents the most updated meta-analysis specifically focused on this important and still commonly overlooked topic in current and future cancer management [9,10,11,12,13,14,15,16]. The pooled results highlighted that the use of adjuvant PD-1 and PD-L1 inhibitors may reduce the risk of relapse compared to patients receiving control treatments (HR, 0.72; 95% CI, 0.67–0.78). In addition, the subgroup analyses observed that this benefit was consistent in different patient populations, including male, female, younger, and older cancer patients. In order to reduce heterogeneity and to increase the statistical power of the current study, the analysis included only PD-1 and PD-L1 inhibitors. The PD-1 pathway represents one of the most important checkpoint pathways; with PD-1, which is expressed on the surface of T cells and binds to its ligands, we are hesitant regarding the inactivation of the T cells’ immune response, and we choose to include only these antagonists targeting PD-L1 or PD-1.

Our meta-analysis presents some strengths and limitations to be noticed. Among the strengths of this study, our analysis included eight controlled clinical trials by using the most updated data in terms of RFS in the intention-to-treat population and in specific subgroups. In addition, we included an overall large number of cancer patients treated with adjuvant immunotherapy. At the same time, some limitations should be underlined. Among these, the current study was based on pooled data, and it was not possible to include single-patient variables. In addition, the studies evaluated different immunotherapeutic agents. All these drugs present different efficacy profiles, an element that could have produced some bias. Moreover, a key point to consider is the inclusion of heterogeneous solid tumors, and the paucity of trials exploring ICIs in each cancer, makes it difficult to draw any reliable conclusions regarding adjuvant immunotherapy in specific cancer types. In fact, due to the diversity of tumor types and adjuvant therapies—as well as the overall limited number of trials—it was not possible to conduct a subgroup analysis on specific cancers. We believe our findings may help guide the everyday treatment decision-making of cancer patients treated with adjuvant PD-1 and PD-L1 inhibitors and assist in the design and interpretation of future studies exploring the role of adjuvant immunotherapy.

## 5. Conclusions

Adjuvant treatment with anti-PD-1 or -PD-L1 is associated with an increased RFS in multiple types of cancer. The results of this meta-analysis confirm the benefit of adjuvant ICIs by taking into account a wide number of patients. Gender and age differences did not affect the RFS benefit; this was confirmed in all subgroups. In future trial design, patients’ selection, which remains the key process to allow these compounds to exert their real value in prolonging survival outcomes, should be improved to individuate specific factors that could implement an immunotherapy response.

## Figures and Tables

**Figure 1 cancers-14-04142-f001:**
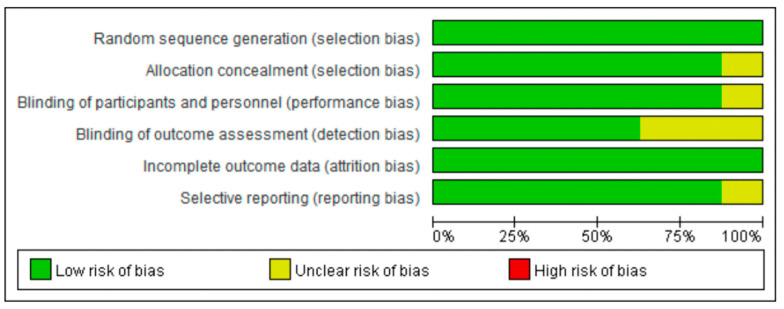
Risk of bias graph, with each risk of bias item reported as a percentage across all included trials.

**Figure 2 cancers-14-04142-f002:**
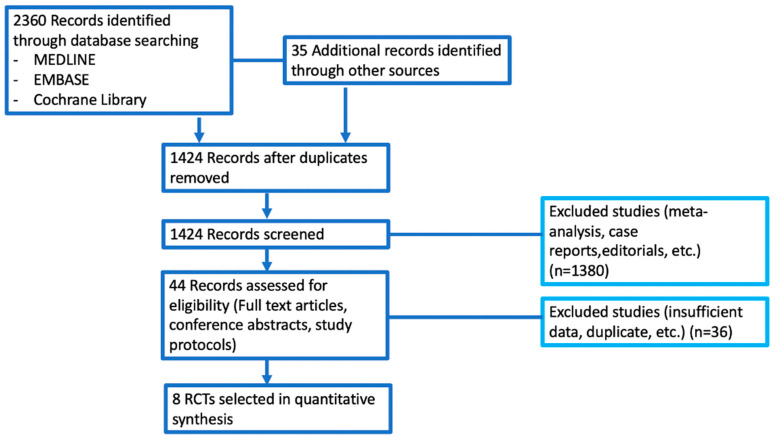
Figure reporting all the trials included and excluded in the quantitative synthesis.

**Figure 3 cancers-14-04142-f003:**
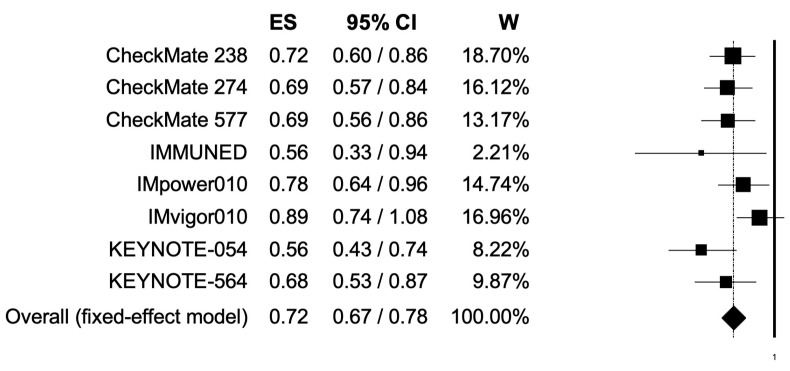
Forest plot of comparison between adjuvant PD-1 and PD-L1 inhibitors versus control (placebo/ipilimumab/observation/best supportive care) in cancer patients; the outcome of interest was relapse-free survival (RFS) in the intention-to-treat population. Abbreviations: CI: confidence interval; ES: effect size; and W: weight.

**Figure 4 cancers-14-04142-f004:**
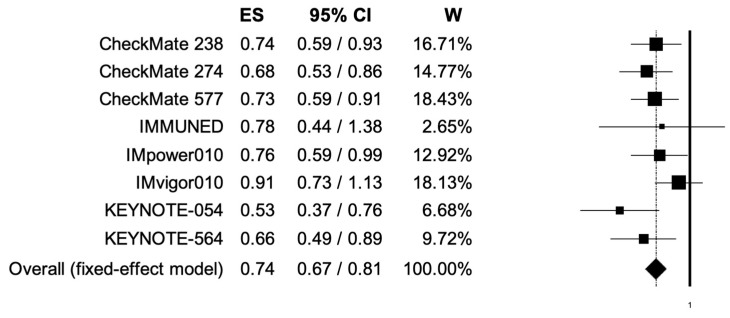
Forest plot of comparison between adjuvant PD-1 and PD-L1 inhibitors versus control (placebo/ipilimumab/observation/best supportive care) in cancer patients; the outcome of interest was relapse-free survival (RFS) in male patients. Abbreviations: CI: confidence interval; ES: effect size; and W: weight.

**Figure 5 cancers-14-04142-f005:**
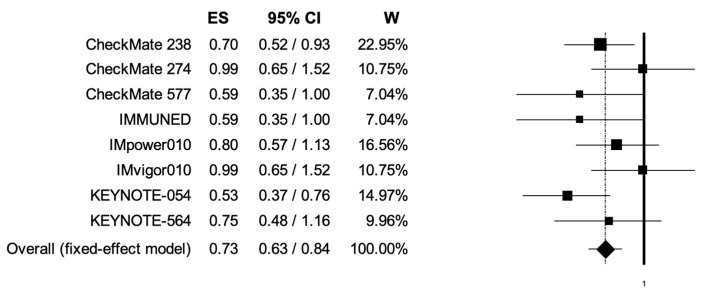
Forest plot of comparison between adjuvant PD-1 and PD-L1 inhibitors versus control (placebo/ipilimumab/observation/best supportive care) in cancer patients; the outcome of interest was relapse-free survival (RFS) in female patients. Abbreviations: CI: confidence interval; ES: effect size; and W: weight.

**Figure 6 cancers-14-04142-f006:**
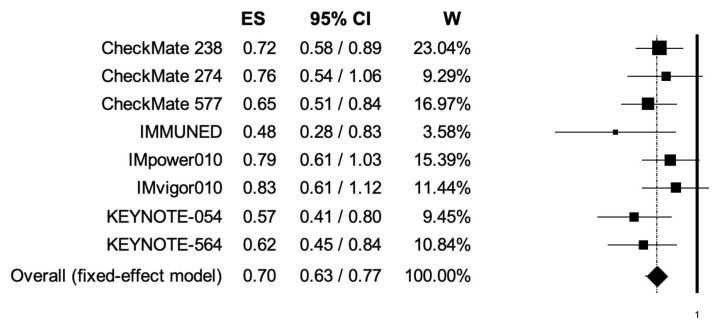
Forest plot of comparison between adjuvant PD-1 and PD-L1 inhibitors versus control (placebo/ipilimumab/observation/best supportive care) in cancer patients; the outcome of interest was relapse-free survival (RFS) in younger patients. Abbreviations: CI: confidence interval; ES: effect size; and W: weight.

**Figure 7 cancers-14-04142-f007:**
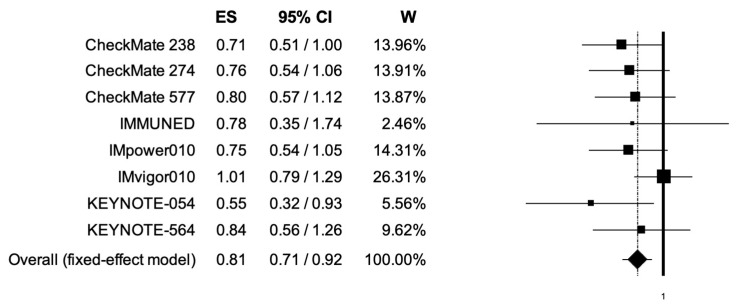
Forest plot of comparison between adjuvant PD-1 and PD-L1 inhibitors versus control (placebo/ipilimumab/observation/best supportive care) in cancer patients; the outcome of interest was relapse-free survival (RFS) in elderly patients. Abbreviations: CI: confidence interval; ES: effect size; and W: weight.

**Table 1 cancers-14-04142-t001:** Summary of the included studies. Abbreviations: RFS: recurrence-free survival; HR: hazard ratio; CI: confidence interval; and ITT: intention-to-treat.

Trial Name (Reference)	Year of Publication	Primary Tumor	Arms Experimental Control	Number of Patients	RFS	HR (CI) *p*
CheckMate 238 [9]	2020	Melanoma	Nivolumab Ipilimumab	906	4-year RFS: 51.7% 41.2%	HR 0.72 (95% CI 0.60–0.86) *p* = 0.0003
KEYNOTE-054 [15]	2021	Melanoma	Pembrolizumab Placebo	1019	1-year RFS ITT: 75.4% 61.0%	HR 0.56 (98.4% CI 0.43–0.74) *p* < 0.0001
IMMUNED [12]	2020	Melanoma	Nivolumab Placebo	167	12.4 months 6.4 months	HR 0.56 (95% CI 0.33–0.94) *p* = 0.011
CheckMate 577 [11]	2021	Esophageal or gastro-esophageal junction cancer	Nivolumab Placebo	794	22.4 months 11.0 months	HR 0.69 (96.4% CI 0.56–0.86) *p* < 0.001
IMpower010 [13]	2021	Non-small cell lung cancer	Atezolizumab Best supportive care	1280	All patients: 42.3 months 35.3 months	HR 0.78 (95% CI 0.64–0.96) *p* = 0.020
IMvigor010 [14]	2021	Urothelial carcinoma	Atezolizumab Observation	809	19.4 months 16.6 months	HR 0.89 (95% CI 0.74–1.08) *p* = 0.24
CheckMate 274 [10]	2021	Urothelial carcinoma	Nivolumab Placebo	709	20.8 months 10.8 months	HR 0.79 (95% CI 0.57–0.84)
KEYNOTE-564 [16]	2021	Renal cell carcinoma	Pembrolizumab Placebo	994	24-months RFS: 77.3% 68.1%	HR 0.68 (95% CI 0.53–0.87) *p* = 0.002

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
