# Peer review of "Adjuvant PD-1 and PD-L1 Inhibitors and Relapse-Free Survival in Cancer Patients: The MOUSEION-04 Study"

_cancers, 2022, doi:10.3390/cancers14174142_

Round 1
Reviewer 1 Report
I believe the manuscript can be accepted in it's current form.
Author Response
Thank you very much for reviewing our manuscript.
Reviewer 2 Report
cancers-1839044
General Comments: This meta-analysis interrogates the impact of ICIs on survival across solid tumors. A few comments:
1) Mild English proofing is needed – a minor point.
2) Table 1 – melanoma, esophageal, and gastroesophageal should not have “-“ in the middle of the words.
3) Can any extrapolations or conclusions be made between different cancer types?
4) Can any extrapolations or conclusions be made between different papers evaluating the same cancer type?
5) Which (if any) of these studies used histologic assessment of IC staining to determine patients that would or would not receive ICI?
Author Response
Dear Reviewer,
Thank you for the time spent revising our paper and for your valuable comments.
- We modified several sentences throughout the manuscript, as suggested.
- Thank you for catching this oversight, that we have now corrected in the text (blue).
- and 4. We better explained this point in the last part of the Discussion, as suggested (red).
- Thank you for this interesting question. The trials of the current analysis included patients regardless of PD-L1 expression, and thus, the analysis includes a heterogeneous, “unselected” patient population.
Thank you again for the time spent revising our paper. We hope the revised manuscript will better suit the journal.
Reviewer 3 Report
In this manuscript, Rizzo, Mollica, and colleagues have exhibited data obtained from a meta analysis of previous studies demonstrating the effects of immune checkpoint inhibitors on altering relapse-free survival in universal cancer patients. The authors have bioinformatically determined the efficacy of the adjuvant PD-1 and PD-L1 inhibitors to diminish clinical relapse risk in different tumors regardless of gender or age differences.
The title sounds less impactive. What is MOUSEION-04? This term should be defined more in detail for general readership if it is intended to be used in the text. Otherwise, “A meta-analysis study reveals relapse-free survival effects of adjuvant PD-1 and PD-L1 inhibitors in cancer patients” can be considered as an alternative.
As frequent targets of immune checkpoints, not only PD-1, PD-L1, and PD-L2 but also CTLA-4 have been studied in antitumor immunotherapy as stated by authors in Introduction. What can be the reasons why the authors focus on quantitative in-silico analysis of previous studies with PD-1 and PD-L1 inhibitors instead of including inhibitors of PD-L2 or CTLA-4? It is recommended to include them in the Introduction.
Current manuscript may have a merit in aspects of exhibiting relapse-free survival effects of the adjuvant PD-1 and PD-L1 inhibitors according to randomized population (Figure 3), genders (Figures 4 and 5) and different aged generations (Figures 6 and 7). Current analysis showed the inhibitors’ relapse-free survival effects in cancer patients in gender- and age-independent manner. It would be also imperative to determine these effects on a few specific delineated and more frequently studied cancer types, via the methodology used in this study with randomized population, which might be further helpful to better systemize anticancer immunotherapeutic effects of the adjuvant PD-1 and PD-L1 inhibitors in cancer type-specific manners.
Comments on data analysis of forest plots shown in Figures 3 to 7: The X-axis of forest plots sounds incomplete. Authors need to complete them. Put the term “Study ID” on the trial names, which can be instrumental to be added.
Comments on Supplemental Figures 1-5 for funnel plots: Some of the legends are invisible. The authors should revise them.
Author Response
Dear Reviewer,
Thank you for the time spent revising our paper and for your valuable comments.
- Thank you for your question. The title MOUSEION refers to a project in which we are involved dedicated to systematically analyze immunotherapy in cancer patients. We already published some works on this topic (PMID: 35031442; PMID: 35029519) and other projects are ongoing. Thus, we would kindly ask to keep the current title, if possible. Thank you for your comprehension.
- Thank you for this suggestion. We decided to focus on PD-1 and PD-L1 inhibitors in order to restrict the “horizon” of the analysis and to try to reduce heterogeneity.
- We further specified the choices behind our analysis in the last part of the Discussion, as suggested (red).
- Thank you for this comment. The program we used (ProMeta software) directly produces the figures in the way you see, and it is not possible to modify them manually.
- We modified the Supplementary material, as suggested.
Thank you again for the time spent revising our paper. We hope the revised manuscript will better suit the journal.
Round 2
Reviewer 2 Report
cancers-1839044-rev1
General Comments: The authors have nicely addressed my comments.
Author Response
Thank you very much for reviewing our manuscript
Reviewer 3 Report
Regarding the response of #2 addressed by the authors, it is recommended to include, those scientific intentions on why quantitative in-silico analysis of previous studies with PD-1 and PD-L1 inhibitors instead of of PD-L2 or CTLA-4 inhibitors was mainly focused, in the Discussion.
Author Response
Dear Reviewer, Thank you for this suggestion. As previously stated, we decided to focus on PD-1 and PD-L1 inhibitors in order to restrict the “horizon” of the analysis and to try to reduce heterogeneity. We better discussed this point in the Discussion section, as suggested. All our changes have been highlighted in red color. Best regards